# Exploring the Novel Dimension of Immune Interactions in Pain: JAK Inhibitors’ Pleiotropic Potential

**DOI:** 10.3390/life13101994

**Published:** 2023-09-29

**Authors:** Krasimir Kraev, Mariela Geneva-Popova, Bozhidar Hristov, Petar Uchikov, Stanislava Popova, Maria Kraeva, Yordanka Basheva-Kraeva, Ivan Sheytanov, Tzvetanka Petranova, Nina Stoyanova, Marin Atanassov

**Affiliations:** 1Department of Propaedeutics of Internal Diseases “Prof. Dr. Anton Mitov”, Faculty of Medicine, Medical University of Plovdiv, 4002 Plovdiv, Bulgaria; 2Clinic of Rheumatology, University Hospital “St. George”, 4000 Plovdiv, Bulgaria; 3Second Department of Internal Diseases, Section “Gastroenterology”, Medical Faculty, Medical University of Plovdiv, 4002 Plovdiv, Bulgaria; 4Clinic of Gastroenterology, University Hospital “Kaspela”, 4001 Plovdiv, Bulgaria; 5Department of Special Surgery, Faculty of Medicine, Medical University of Plovdiv, 4002 Plovdiv, Bulgaria; 6Second Surgery Clinic, University Hospital “St. George”, 4000 Plovdiv, Bulgaria; 7Department of Otorhinolaryngology, Medical Faculty, Medical University of Plovdiv, 4002 Plovdiv, Bulgaria; 8Department of Ophthalmology, Faculty of Medicine, Medical University of Plovdiv, University Eye Clinic, University Hospital, 4000 Plovdiv, Bulgariamarin_aa@abv.bg (M.A.); 9Department of Rheumatology, Clinic of Rheumatology, University Hospital St. Ivan Rilski, Medical Faculty, Medical University of Sofia, 1000 Sofia, Bulgaria

**Keywords:** neurobiology of pain, JAK inhibitors

## Abstract

This review explores the link between immune interactions and chronic pain, offering new perspectives on treatment. It focuses on Janus kinase (JAK) inhibitors’ potential in pain management. Immune cells’ communication with neurons shapes neuroinflammatory responses, and JAK inhibitors’ effects on pain pathways are discussed, including cytokine suppression and microglial modulation. This review integrates studies from rheumatoid arthritis (RA) pain and central sensitization to highlight connections between immune interactions and pain. Studies on RA joint pain reveal the shift from cytokines to sensitization. Neurobiological investigations into central sensitization uncover shared pathways in chronic pain. Clinical evidence supports JAK inhibitors’ efficacy on pain-related outcomes and their effects on neurons and immune cells. Challenges and future directions are outlined, including interdisciplinary collaboration and dosing optimization. Overall, this review highlights JAK inhibitors’ potential to target immune-mediated pain pathways, underscoring the need for more research on immune–pain connections.

## 1. Introduction

Pain, a fundamental aspect of human experience, serves as an alarm system that alerts us to potential threats and harm. While acute pain is a crucial protective mechanism, chronic pain represents a complex, debilitating condition that often lacks effective treatment options [1]. Chronic pain conditions, such as neuropathic pain and inflammatory pain, pose significant challenges due to their multifaceted etiology and variable response to traditional analgesic interventions [1,2,3].

Recent advancements in our understanding of the immune system’s involvement in pain processing have opened new avenues for research and therapeutic exploration [4,5]. The immune system, traditionally recognized for its role in defending the body against infections, has emerged as a critical player in shaping pain perception and modulation. In parallel, pharmaceutical interventions targeting the Janus kinase (JAK) signaling pathway, originally designed to modify immune responses in autoimmune diseases, have shown unexpected pleiotropic effects that extend to pain modulation [6,7].

## 2. Nociceptor Sensory Neuron–Immune Interactions

In a seminal work by Pinho-Ribeiro et al. [8], the intricate interactions between nociceptor sensory neurons and the immune system in the context of pain and inflammation are meticulously explored. This research delves into the indispensable role of the immune system in pain sensation, as it releases molecular mediators that sensitize nociceptor neurons. This sensitization is closely linked to tissue injury and inflammation, marking the interplay between immune responses and pain processing.

The receptors and ion channels present in nociceptor peripheral nerve terminals act as sentinels, detecting chemical mediators released during inflammation. These mediators serve as signals that initiate an action potential, which is subsequently transmitted to nociceptor cell bodies in the dorsal root ganglia (DRG) [9,10]. From there, the signal continues its journey to the spinal cord and brain, where it is interpreted as pain. During inflammation, the threshold for nociceptor neurons to produce action potentials is lowered, leading to heightened pain sensitivity or “hyperalgesia”. This phenomenon plays a crucial role in chronic pain associated with inflammatory disorders like rheumatoid arthritis and inflammatory bowel disease.

Recent research efforts have focused on identifying specific immune cells and mediators implicated in chronic pain. The pivotal role of the immune system in neural sensitization becomes evident, as pain tends to decrease once the tissue immune response resolves. This underscores the relevance of immune–neural pathways in chronic pain conditions, paving the way for potential remedies.

Nociceptor neurons express a diverse array of receptors for various immune mediators, including cytokines, lipids, proteases, and growth factors generated by immune cells. Upon activation of these receptors, intricate signaling cascades are initiated. These cascades induce changes in the gating characteristics of ion channels such as TRPV1, TRPA1, Nav1.7, Nav1.8, and Nav1.9 through processes like phosphorylation [11,12]. The outcome of these changes is enhanced neuronal firing, contributing to heightened pain sensitivity.

Emerging research is beginning to illuminate the roles of distinct immune mediators in mediating pain sensitivity across different illness contexts. For example, studies involving mice models of carrageenan-induced inflammatory pain and neuropathic pain have shown that neutrophils contribute to pain maintenance through the production of cytokines and prostaglandin E2 (PGE2). Additionally, non-neutrophilic CD11b+ myeloid cells, which are presumed to be macrophages, are implicated in pain sensitization following incisional wound damage [13].

Mast cells also emerge as key players in the sensitization of nociceptors. Detailed examination through electron microscopy reveals a strong relationship between nociceptor nerve terminals and mast cells, particularly in mucosal tissues. Upon activation, mast cells release an array of pain-sensitizing molecules, including cytokines (IL-5, TNF, IL-6, IL-1), histamine, 5-HT, and nerve growth factor (NGF) [14]. This release triggers pain sensitization by interacting with receptors present on nociceptors. Importantly, mast cells do not only contribute to pain during acute inflammation but also aggregate in chronic inflammatory diseases, where they further contribute to pain chronicity.

Macrophages, being sentinel myeloid cells found throughout the body, and monocytes, blood-borne myeloid cells drawn to inflammatory sites upon tissue injury, are extensively implicated in chronic pain conditions. These cells produce a plethora of inflammatory cytokines, growth factors, and lipids that can directly act on nociceptor neurons, contributing to the sensation of pain.

Interestingly, research suggests that immune cells might interact not only at the site of injury but also with the cell bodies of nociceptor neurons within the dorsal root ganglia (DRGs) [15]. The DRG soma serves as a control center for neuroplasticity and long-term sensitization through protein synthesis. Evidence indicates that in conditions of chronic pain, the number of innate and adaptive immune cells within the DRGs increases [16]. Studies involving chemotherapy and sciatic-nerve-ligation-induced neuropathic pain reveal elevated levels of macrophages, monocytes, neutrophils, and T cells in the DRGs. In fact, T cells have been found to release leukocyte elastase in the DRGs, contributing to pain after nerve injury. Additionally, activated mast cells in the DRGs have been associated with pain in sickle cell anemia. By unraveling the intricate crosstalk between immune cells and neurons, this study uncovers the intricate cellular and molecular mechanisms that drive pain sensitization and hypersensitivity.

## 3. Central Sensitization and Neurobiology of Pain

Central sensitization, a phenomenon intricately linked to the development and persistence of chronic pain conditions, is thoroughly investigated by Harte et al. [17]. This exploration unveils the molecular and cellular underpinnings of central sensitization, offering valuable insights into the convergence of immune and neural pathways that perpetuate pain hypersensitivity.

Central sensitization represents a pivotal mechanism through which the nervous system amplifies pain signals, contributing to the intensification of pain perception. Harte’s research meticulously dissects the intricate neurobiological processes that underlie this phenomenon [17]. By delving into the molecular interactions and cellular changes that fuel central sensitization, Harte’s study uncovers the intricate web of immune–neural interactions that drive the maintenance of chronic pain.

The convergence of immune responses and neural pathways becomes evident in the context of central sensitization. Immune cells, particularly microglia and macrophages, play a crucial role in mediating neuroinflammation, which is a hallmark of central sensitization. These cells release a cascade of cytokines, chemokines, and growth factors that shape the neuroinflammatory microenvironment, contributing to the heightened excitability of pain pathways.

Moreover, the neurobiological changes observed in central sensitization involve alterations in synaptic plasticity and neurotransmitter release, further underscoring the intricate relationship between immune responses and pain perception. The insights gleaned from Harte’s study emphasize the need to consider both immune and neural factors in developing targeted interventions for chronic pain conditions [17].

In conclusion, the comprehensive exploration of pain, immune interactions, and central sensitization highlights the intricate interplay between the immune and nervous systems in shaping pain perception and modulation. From the fundamental role of pain as an alarm system to the complexities of chronic pain conditions, the insights from these studies underscore the need for interdisciplinary approaches in understanding and treating pain. The discoveries of immune–neural interactions and their roles in nociceptor sensitization and central sensitization provide potential avenues for the development of innovative therapies that address the multifaceted nature of chronic pain. As the fields of immunology and neuroscience converge, there is hope for breakthroughs that will bring relief to those suffering from the burden of chronic pain.

## 4. Pain in the Background of Rheumatic Diseases

Pain is an intricate and universal facet of human experience, acting as a vigilant sentinel that warns of impending harm and danger. While acute pain is a fundamental protective mechanism, chronic pain represents a complex and often debilitating condition that poses significant challenges for effective management. The realm of rheumatic diseases, encompassing disorders such as rheumatoid arthritis (RA), ankylosing spondylitis, and psoriatic arthritis, is characterized by the coexistence of joint inflammation and a substantial burden of pain [18]. This review delves into the intricate landscape of pain within the context of rheumatic diseases, exploring its underlying mechanisms, impact, and potential avenues for improved management.

## 5. The Complexity of Pain in Rheumatic Diseases

Pain in rheumatic diseases transcends the boundaries of a singular entity; it is a multifaceted phenomenon intertwined with the intricate interactions between immune responses and pain-signaling pathways [19]. In conditions like rheumatoid arthritis, ankylosing spondylitis, and psoriatic arthritis, the experience of pain is not solely a result of structural damage but a dynamic interplay between immune activation, neuroinflammatory processes, and altered pain perception. This complexity necessitates a comprehensive understanding of the underlying mechanisms to address the full spectrum of pain experiences in these diseases.

## 6. The Role of Immune Interactions

Recent advancements in immunology have shed light on the pivotal role of immune interactions in shaping pain perception in the context of rheumatic diseases. Immune cells, including microglia, macrophages, and T cells, infiltrate pain-relevant regions such as the spinal cord and dorsal root ganglia in response to injury and inflammation. Within these regions, these immune cells release an array of proinflammatory and anti-inflammatory cytokines, chemokines, and growth factors, contributing to the establishment of a complex neuroinflammatory microenvironment [20]. This intricate milieu of signaling networks actively participates in the modulation of pain signaling cascades, thereby contributing to the heightened pain sensitivity observed in rheumatic diseases [21].

## 7. Bidirectional Communication

The communication between immune cells and neurons within the realm of pain is a dynamic and bidirectional process [16,17]. Neurons express receptors for immune mediators, enabling them to sense and respond to cytokines and other immune signaling molecules. Conversely, immune cells exhibit a heightened sensitivity to neuronal signaling, allowing them to adjust their activation states and cytokine release patterns accordingly [20,21]. This intricate interplay creates a fertile ground for neuroinflammation, a key driver of chronic pain conditions in rheumatic diseases. As neuroinflammation amplifies pain signaling, a self-sustaining loop is established, further contributing to the perpetuation of chronic pain [16,17].

As we navigate the complex terrain of pain in rheumatic diseases, a myriad of challenges and opportunities lie on the horizon [18,19]. Given the diverse nature of chronic pain experiences, optimizing interventions tailored to specific pain conditions and patient profiles is imperative. Interdisciplinary collaboration between immunologists, neuroscientists, and pain researchers is paramount in untangling the intricacies of immune–neural crosstalk and identifying novel therapeutic targets. Rigorous clinical trials, focusing on pain-specific endpoints, will provide direct evidence of their impact on pain perception, thus guiding the development of personalized and effective pain management strategies [20].

The landscape of pain within the context of rheumatic diseases is a tapestry woven from the intricate interactions between immune responses and pain-signaling pathways. The involvement of the immune system in pain processing, particularly evident in diseases like rheumatoid arthritis, underscores the complexity of pain perception. As researchers delve deeper into the mechanisms governing immune–neural interactions, they pave the way for innovative pain management strategies within the realm of rheumatic diseases [18,19]. This interdisciplinary approach holds the potential to enhance patients’ quality of life by specifically targeting the intricate connections between immune responses and pain perception. With each discovery, the prospect of transformative advancements in rheumatic pain management becomes increasingly tangible, renewing hope for better days for those burdened by chronic pain [22,23].

## 8. The Multifaceted Nature of Pain in Rheumatoid Arthritis

Pain in rheumatoid arthritis is far from a one-dimensional experience. It presents as a complex interplay of nociceptive, inflammatory, and neuropathic components. Nociceptive pain arises from the activation of peripheral nociceptors in response to tissue damage and inflammation within affected joints [18]. This is compounded by inflammatory pain, a result of the release of proinflammatory cytokines such as tumor necrosis factor-alpha (TNF-α) and interleukin-6 (IL-6), which sensitize pain receptors and contribute to the persistence of pain. Neuropathic pain, characterized by shooting, burning, or electric sensations, emerges due to nerve damage caused by chronic inflammation [20,24]. 

## 9. Cytokines in Rheumatoid Arthritis (RA) Pain Mechanisms

Cytokines play a central role in the genesis of pain associated with rheumatoid arthritis (RA) [25]. RA is characterized by persistent joint inflammation and debilitating pain, and several cytokines have been implicated in its pathogenesis, including TNF-α, IL-1, and IL-6 (Figure 1). These cytokines contribute to RA and its associated pain through the promotion of autoimmunity, chronic inflammatory synovitis, and the destruction of nearby joint tissues [26].

Other cytokines, such as granulocyte–macrophage colony-stimulating factor (GM-CSF), IL-4, IL-10, IL-13, and IL-17, also influence pain and inflammation in RA [27,28,29,30,31,32,33].

GM-CSF, for instance, signals through various pathways, including the JAK2/STAT5 pathway. It stimulates the release of proinflammatory cytokines and chemokines like CCL17, which can directly and indirectly induce nociception [34]. GM-CSF is associated with pain in conditions like bone cancer, inflammation, and arthritis, and it is detectable in the synovial fluid of RA patients [31,35,36].

IL-1β, a proinflammatory cytokine, is prominently upregulated in RA and the synovial milieu. It is a major contributor to inflammation and structural damage in arthritis and can also recruit and activate T cells, contributing to hyperalgesia [37].

In contrast, anti-inflammatory cytokines like IL-4, IL-10, and IL-13 have pain-alleviating properties [32]. However, these cytokines also signal directly through the JAK/STAT pathway, highlighting the delicate balance between pronociceptive and antinociceptive cytokines and their combined impact on pain responses.

IL-6, a proinflammatory cytokine and a principal mediator of systemic inflammation in RA, directly activates the JAK/STAT3 pathway. It plays a significant role in regulating joint pain. IL-6 affects pain both peripherally and at the dorsal root ganglion (DRG). It exerts its effects through the binding of sIL-6R and trans-signaling via gp130, present in various components of the nociceptive system, including neurons, glial cells, and DRGs. Emerging findings suggest that IL-6 family cytokines may also influence neuronal survival, differentiation, and regeneration [38,39]. Moreover, IL-6 has been linked to pain in various other conditions, including bone cancer, spinal cord or peripheral nerve injury, and neuropathy due to chemotherapy [40].

IL-17A, another proinflammatory cytokine, is elevated in the synovial fluid of RA patients. In rodent models, IL-17 injections induce hyperalgesia, while mice lacking IL-17A show reduced mechanical hyperalgesia after inflammation induction [41]. Although the exact mechanism remains unclear, IL-17A may signal through the JAK/STAT pathway, potentially modulating neuronal activity through VEGF [42]. IL-17 receptors on sensory neurons further suggest a role in nociception [43].

TNF-α, binding to TNF receptor 1 (TNFR1) and TNF receptor 2 (TNFR2), has been detected in dorsal root ganglia (DRGs) in animal studies. TNF-α binding to these receptors contributes to hyperalgesia in chronic inflammation, affecting various parts of the nervous system, including the spinal cord, thalamus, periaqueductal gray, and amygdala. It has also been implicated in neuropathic pain, with increased expression of TNFR1, TNFR2, and TNF-α after peripheral nerve injury [44,45].

**Figure 1 life-13-01994-f001:**
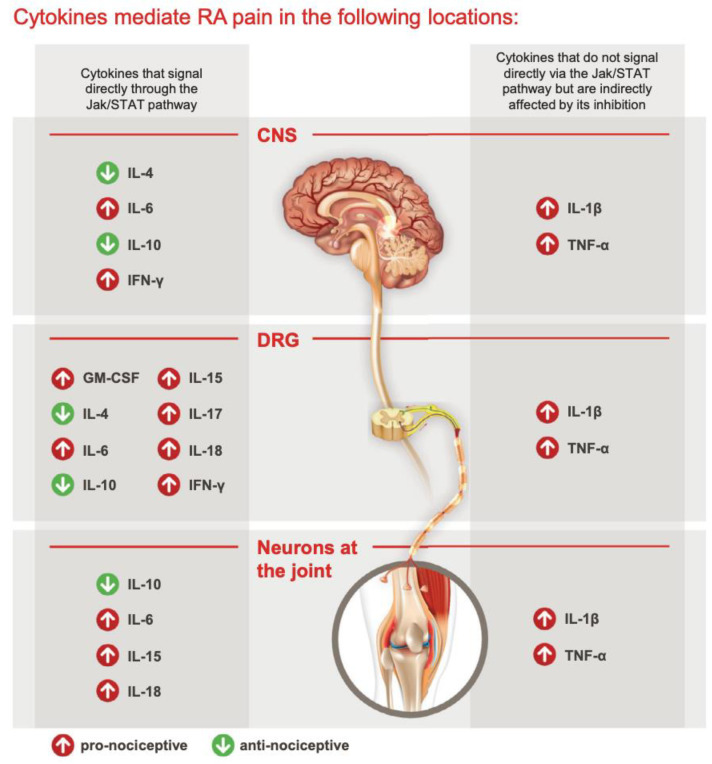
Cytokines and RA pain by Simon, L.S. et al. (2021). The JAK/STAT Pathway: A Focus on Pain in Rheumatoid Arthritis. Seminars in Arthritis and Rheumatism [46].

## 10. Peripheral Sensitization and Immune Interactions

The immune system plays a pivotal role in the pain experienced by RA patients. Immune cells infiltrate the synovium, releasing a barrage of inflammatory mediators that sensitize pain receptors [19,33]. The interaction between nociceptor sensory neurons and immune cells creates a feedback loop that amplifies pain perception. Immune cells release cytokines and other signaling molecules that activate pain receptors, while the nerve terminals of nociceptors express receptors for these immune mediators, further heightening pain sensitivity [34]. Mast cells, neutrophils, macrophages, and T cells all contribute to this neuro–immune crosstalk, perpetuating pain [36].

## 11. Central Sensitization and Neuroplasticity

The complexities of pain perception in RA extend beyond peripheral sensitization. Central sensitization, a phenomenon involving heightened excitability of neurons in the spinal cord and brain, plays a pivotal role [16,17]. Persistent peripheral nociceptive input leads to adaptive changes in the central nervous system, amplifying pain signals and causing even innocuous stimuli to be perceived as painful. Neuroplasticity, the brain’s ability to rewire itself in response to chronic pain, further contributes to the long-lasting nature of pain in RA [20,21].

## 12. Clinical Implications and Pain Management Strategies

Understanding the multifaceted nature of pain in RA has significant clinical implications. Traditional analgesics and nonsteroidal anti-inflammatory drugs (NSAIDs) offer relief by targeting peripheral pain pathways, but their efficacy in managing chronic pain is limited [20]. Targeted biologic therapies, such as TNF inhibitors and IL-6 receptor antagonists, have revolutionized RA treatment by curbing inflammation and alleviating pain. Additionally, disease-modifying antirheumatic drugs (DMARDs) slow down joint damage and can indirectly improve pain. Emerging treatments, including nerve growth factor (NGF) inhibitors, hold promise in directly targeting pain pathways [23].

## 13. Can We Achieve a Patient-Centered Approach?

It is crucial to recognize the impact of pain on RA patients’ lives. Beyond the physical toll, pain affects emotional well-being, social interactions, and daily activities. A multidisciplinary approach encompassing pharmacological interventions, physical therapy, psychological support, and patient education is necessary to address pain comprehensively [18,19].

Pain in rheumatoid arthritis is a multidimensional phenomenon driven by intricate interactions between immune responses, nerve pathways, and central sensitization. Understanding the underlying mechanisms is pivotal in devising effective pain management strategies that can enhance the quality of life of individuals living with RA. As research continues to unravel the complexities of pain, the hope is to usher in a new era of personalized pain management that targets the unique pain profiles of RA patients [19,20].

## 14. JAK/STAT Signaling Pathway and Its Implications in Pain Modulation

The Janus kinase (JAK) signaling pathway, intricately intertwined with the signal transducer and activator of transcription (STAT) pathway, holds a pivotal role in orchestrating cellular responses to a variety of extracellular signals (Figure 2). This complex and versatile pathway is not only involved in immune responses but also plays a critical role in the intricate network of neural processes, including pain modulation. The JAK family, comprised of distinct isoforms such as JAK1, JAK2, JAK3, and TYK2, collaborates with various STAT proteins to transduce signals from cytokine and growth factor receptors to the nucleus, influencing gene expression and cellular behavior (Figure 3). This intricate signaling cascade not only regulates immune cell activation, proliferation, and differentiation but also extends its reach to neural signaling pathways, bridging the gap between the immune system and the nervous system [22,23].

**Figure 2 life-13-01994-f002:**
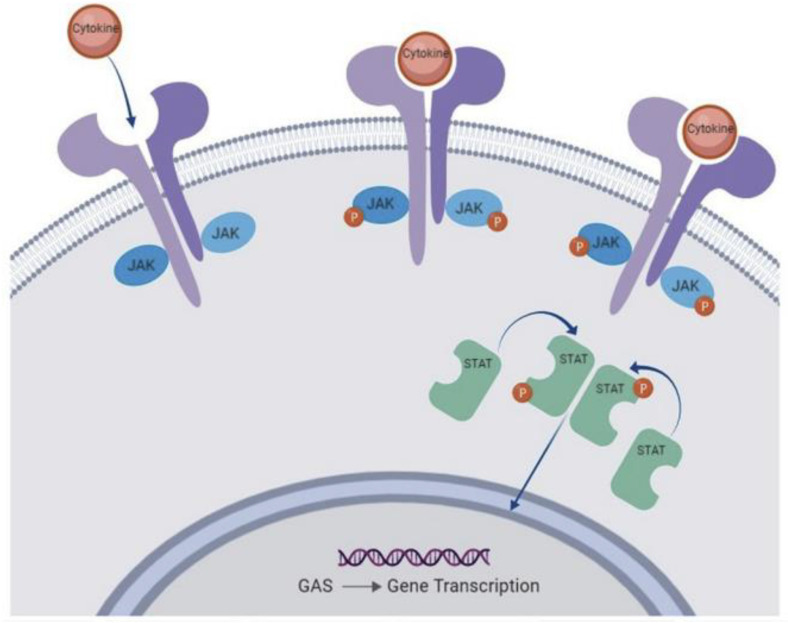
The JAK/STAT signaling pathway; K. Bechman et al. The new entries in the therapeutic armamentarium: The small molecule JAK inhibitors, Pharmacological Research, Volume 147, 2019, 104392, ISSN 1043-6618, https://doi.org/10.1016/j.phrs.2019.104392 [47].

**Figure 3 life-13-01994-f003:**
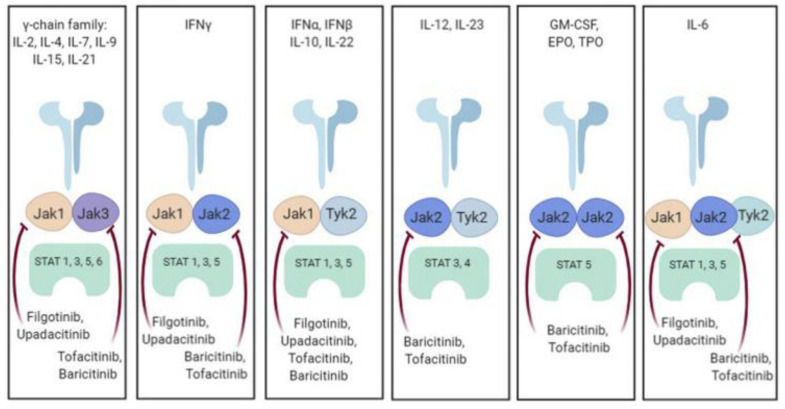
Cytokine signaling through JAK/STAT pathway; K. Bechman et al. The new entries in the therapeutic armamentarium: The small molecule JAK inhibitors, Pharmacological Research, Volume 147, 2019, 104392, ISSN 1043-6618, https://doi.org/10.1016/j.phrs.2019.104392 [47].

In the context of pain modulation, the JAK/STAT signaling pathway emerges as a crossroads where immune responses intersect with neural processes, contributing to the complexity of pain perception. Neuro–immune interactions become particularly pertinent as JAK/STAT signaling influences the activation of various immune cells that infiltrate pain-relevant regions, such as microglia, macrophages, and T cells. These immune cells, once activated, release an array of cytokines, chemokines, and growth factors that collectively shape the neuroinflammatory microenvironment. This interplay between immune responses and neural pathways contributes to the development, maintenance, and amplification of chronic pain conditions, serving as a foundation for the bidirectional communication that characterizes pain hypersensitivity [22,23,27].

## 15. JAK Inhibitors: A New Class of Drugs

Janus kinase inhibitors, commonly referred to as JAK inhibitors, represent a promising class of drugs with applications in the treatment of various medical conditions [26,27,28]. These inhibitors target Janus kinases, a family of enzymes that play a crucial role in transmitting signals for cell growth and immune response. By modulating JAK activity, these inhibitors have emerged as valuable therapeutic tools for conditions ranging from autoimmune diseases to certain cancers. In this article, we explore the significance of JAK inhibitors in modern medicine, their mechanism of action, and their current and potential applications [26,27].

Janus kinases are essential components of cell signaling pathways, particularly those involving cytokines, which are small proteins that regulate immune responses and inflammation. When these pathways become dysregulated, they can contribute to autoimmune diseases and certain cancers. JAK inhibitors work by interfering with these signaling pathways. They bind to the Janus kinases, inhibiting their activity and thus reducing the downstream effects of cytokines [24]. This mechanism helps control inflammation and suppresses the immune system, making JAK inhibitors effective in managing autoimmune disorders [25].

JAK inhibitors have shown remarkable success in treating autoimmune diseases such as rheumatoid arthritis, psoriasis, and inflammatory bowel disease. Medications like Tofacitinib (trade name: Xeljanz), Upadacitinib (trade name: Rinvoq), and Baricitinib (trade name: Olumiant) are among the JAK inhibitors approved for these conditions (Table 1) [25]. By targeting the overactive immune responses characteristic of autoimmune diseases, JAK inhibitors provide relief to patients and improve their quality of life.

In addition to autoimmune diseases, JAK inhibitors have demonstrated potential in cancer therapy. They are being investigated as targeted treatments for various malignancies, including myelofibrosis, a rare bone marrow disorder. Ruxolitinib (trade name: Jakafi) and Fedratinib (trade name: Inrebic) are examples of JAK inhibitors used in the treatment of myelofibrosis [27,31]. These inhibitors help control abnormal cell proliferation and alleviate disease-related symptoms.

### Recent Safety Concerns

While JAK inhibitors offer significant therapeutic benefits, recent safety concerns have emerged. Regulatory agencies like the FDA and EMA have issued warnings regarding potential risks, including serious heart-related events, cancer, blood clots, and death, associated with the use of JAK inhibitors [28,29,30,31]. These concerns underscore the importance of ongoing research and monitoring to better understand the long-term safety profile of these drugs.

## 16. Role of JAK Inhibitors in Modulating Pain Perception

JAK inhibitors, initially conceptualized as immune-modulating agents, have unveiled a multifaceted impact that reaches beyond the immune system, encompassing neural and glial cells, particularly microglia, which play a central role in neuroinflammation and the perpetuation of chronic pain states. This expanded scope of JAK inhibitors provides an intriguing avenue for pain modulation, tapping into the intricate interplay between immune responses and neural pathways.

One of the key mechanisms through which JAK inhibitors impact pain perception involves the suppression of proinflammatory cytokines, such as interleukin-1β (IL-1β), interleukin-6 (IL-6), and tumor necrosis factor-alpha (TNF-α). By interrupting JAK-mediated signaling pathways essential for cytokine production, JAK inhibitors could potentially dampen the inflammatory response that underlies neuroinflammation and hypersensitivity. This suppression of proinflammatory mediators might contribute to the alleviation of pain-related behaviors and the restoration of neural homeostasis [22,23].

Furthermore, JAK inhibitors have the potential to restore a balance between neuroinflammation and anti-inflammatory pathways. By promoting the production of anti-inflammatory cytokines, like interleukin-10 (IL-10), JAK inhibitors could counteract the detrimental effects of proinflammatory cytokines. This rebalancing might mitigate the hypersensitivity associated with chronic pain conditions, providing a dynamic modulation of pain perception [23].

Experimental evidence from animal models of chronic pain, coupled with clinical trials primarily conducted in autoimmune diseases, underscores the potential of JAK inhibitors as promising pain modulators. These inhibitors have demonstrated efficacy in attenuating pain-related behaviors and improving pain outcomes in patients. However, to fully exploit the potential of JAK inhibitors in pain management, dedicated clinical trials focusing specifically on pain conditions are warranted. The landscape of pain research is rapidly evolving as scientists delve deeper into the molecular and cellular mechanisms linking JAK/STAT signaling with pain modulation [22,23,46].

## 17. Translating Bench to Bedside: Challenges and Prospects

The journey from laboratory insights to effective pain management strategies using JAK inhibitors faces several challenges. The intricacy of the JAK/STAT pathway demands a nuanced understanding of its diverse effects on different cell types and tissues. As researchers strive to develop selective JAK inhibitors that target specific isoforms, the risk of off-target effects and unintended consequences remains a concern. Balancing the suppression of proinflammatory signals with the preservation of vital immune responses poses a delicate tightrope walk in therapeutic development.

Clinical translation also necessitates a comprehensive grasp of the varying etiologies of chronic pain conditions. Different pain conditions, such as neuropathic pain, inflammatory pain, and central sensitization, may involve distinct molecular and cellular mechanisms. Tailoring JAK-inhibitor-based therapies to specific pain conditions requires a personalized approach accounting for the heterogeneity of patients’ experiences.

Furthermore, long-term safety profiles of JAK inhibitors, especially in the context of chronic pain management, warrant rigorous investigation. The effects of sustained JAK inhibition on immune surveillance, infection susceptibility, and overall homeostasis necessitate meticulous monitoring. Additionally, potential interactions with other pain medications and their impact on the delicate balance of neural and immune responses need to be elucidated.

## 18. The Future Landscape of Pain Management

As the realm of pain management undergoes a paradigm shift, the convergence of JAK/STAT signaling with immune–neural interactions unveils a compelling terrain for exploring innovative strategies. The unexpected role of JAK inhibitors in neural pathways, beyond their initial immune-modulating context, offers a promising avenue for alleviating chronic pain conditions. As researchers unravel the intricate mechanisms through which JAK inhibitors influence pain perception, interdisciplinary collaborations between immunologists, neuroscientists, and clinicians will be vital to harnessing the full potential of these agents. Ultimately, the dynamic interplay between immune responses and neural processes holds the key to transforming pain modulation and enhancing the lives of individuals grappling with the intricate challenges of chronic pain. Through the lens of JAK/STAT signaling, the bridge between the immune and nervous systems illuminates new avenues of hope and relief for those burdened by the persistent weight of pain.

## 19. Discussion

The exploration of pain within the contexts of rheumatic diseases as a whole, rheumatoid arthritis itself, immune interactions, and the JAK/STAT signaling pathway reveals a fascinating interplay between the immune and nervous systems that shapes pain perception and modulation. This intricate relationship has significant implications for our understanding of chronic pain and the development of novel pain management strategies.

The text emphasizes the multifaceted nature of pain experiences in conditions like rheumatoid arthritis, where pain is not solely a result of structural damage but rather a complex interplay between immune responses and altered pain perception. The integration of immune cells, such as microglia, macrophages, and T cells, within pain-relevant regions highlights the importance of neuro–immune crosstalk in perpetuating chronic pain states. The immune system’s role in sensitizing pain receptors and shaping the neuroinflammatory microenvironment establishes the foundation for heightened pain sensitivity.

A noteworthy highlight is the potential of Janus kinase (JAK) inhibitors as therapeutic agents for pain modulation. Originally designed to target immune pathways, these inhibitors have demonstrated unexpected effects on pain pathways, extending their impact beyond immune responses. By suppressing proinflammatory cytokines and promoting a balance between pro- and anti-inflammatory signals, JAK inhibitors offer a promising approach to addressing chronic pain. This highlights the interconnectedness between immune and neural processes and opens the door to innovative pain management strategies.

The discussion also addresses challenges in translating these findings into effective clinical applications. Balancing the selective inhibition of JAK isoforms to minimize off-target effects and understanding the diverse mechanisms underlying different pain conditions are crucial steps. Additionally, long-term safety profiles and potential interactions with existing pain medications require thorough investigation.

In conclusion, this text underscores the importance of interdisciplinary collaboration in unraveling the complexities of pain modulation. The convergence of immunology and neuroscience offers exciting prospects for developing personalized pain management strategies that consider the intricate connections between immune responses and pain perception. As our understanding of these interactions deepens, there is hope for transformative breakthroughs that will alleviate the burden of chronic pain and enhance the well-being of individuals living with these conditions.

## 20. Conclusions

The exploration of pain within the contexts of rheumatic diseases, immune interactions, and the JAK/STAT signaling pathway reveals the intricate and dynamic interplay between immune responses and neural processes that shape pain perception and modulation. From the complex nature of pain experiences in conditions like rheumatoid arthritis to the bidirectional communication between immune cells and neurons, a comprehensive understanding of these interactions is crucial for developing effective pain management strategies. The emergence of JAK inhibitors as potential pain modulators provides a promising avenue for addressing chronic pain conditions by targeting both immune-mediated and neural pathways. While challenges in clinical translation and safety considerations persist, the potential of JAK inhibitors in transforming the landscape of pain management is undeniable. Through interdisciplinary collaborations and continued research, the bridge between immune responses and neural processes holds the promise of alleviating the burden of chronic pain and improving the quality of life for individuals living with these conditions. As science advances, so too does the hope for innovative therapies that unravel the complexities of pain and offer relief to those who suffer.

## Figures and Tables

**Table 1 life-13-01994-t001:** JAK inhibitors commercially available in Europe.

Trade Names	Janus Kinase Inhibitors
Xeljanz (Pfizer)	Tofacitinib
Rinvoq (Abbvie)	Upadacitinib
Olumiant (Elli Lilly)	Baricitinib
Cibinqo (Pfizer)	Abrocitinib
Jyseleca (Gilead)	Filgotinib

## Data Availability

Not applicable.

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
