# Peer review of "Exploring the Novel Dimension of Immune Interactions in Pain: JAK Inhibitors’ Pleiotropic Potential"

_life, 2023, doi:10.3390/life13101994_

Round 1

Reviewer 1 Report

The authors try to do an interesting paper with some information regarding pain in general and the possible role of JAK on it.

The paper idea is good, but I’m afraid to say that the manuscript lacks on bibliography references.

The introduction section don’t have even one reference to corroborate with the text, and the other topics you should explore more the literature data to make the text more strong. Only 20 references for a review paper is not enough.

The topics should also be numbered.

Author Response

Dear reviewer, first of all I want to thank you for your fruitful review, hopefully my corrections will satisfy your demands and my article will be accepted for publication. 
This is my point-by-point response:

"The introduction section don’t have even one reference to corroborate with the text, and the other topics you should explore more the literature data to make the text more strong. Only 20 references for a review paper is not enough." 

Answer: I have added new references, numbered them in the text as well, now I have 48 references.

The topics should also be numbered.

Answer: I have numbered the topics.

All the corrections for the current reviewer are highlighted in red for easier check.

Reviewer 2 Report

The review by Kasimir Kraev is well-written and addresses a very important topic such as the mechanisms underlying chronic pain, particularly pain arising from rheumatic conditions such as rheumatoid arthritis.

The work focuses on the involvement of the immune system and the role of the JAK/STAT pathway, specifically on the pleiotropic potential of JAK inhibitors in pain.

However, the review does not appear to be thoroughly explored in all its parts. It contains numerous repetitions and does not add much to what is already present in literature. Please, for example, consider the following publication      PMID: 33412435.

Furthermore, I find it lacking that  JAK inhibitors are not even mentioned or explored in depth.

In conclusion, the way this review is presented appears to provide more informative value for a general audience rather than for the scientific community.  

Adequate 

Author Response

Dear reviewer, thank you for your honest and fruitful review, I have made the required changes and I hope they will satisfy your demands so my article can be accepted for publication.

This is my point-by-point response 

" The review does not appear to be thoroughly explored in all its parts. It contains numerous repetitions and does not add much to what is already present in literature. Please, for example, consider the following publication      PMID: 33412435."

I have expanded the review, I have added new information, using the article you recommended and I really want to thank you for this article.

Furthermore, I find it lacking that  JAK inhibitors are not even mentioned or explored in depth.

I have added a couple of paragraphs about JAK inhibitors even information regarding new concerns about them.

All the changes I have made because of your reviews are marked with green.

Reviewer 3 Report

The manuscript by Kraev nicely reviews the involvement of JAK in the modulation of pain and its clinical importance for the future treatment of pain. I feel this may be welcomed by the journal's audience, but I have several comments as below.

Specific comment

1.     Since some paragraph repeats the similar message (e.g., lines 132-149 and lines 192-197), authors had better revise the manuscript by reconfiguration.

2.     Authors should add appropriate references in the text. For example, there are no references in the text lines 162-190, lines 207-223).

3.     Authors had better add some figures or schemes that explain the JAK-mediated mechanism of pain for example for easier understanding for the reader.

4.     Although authors focused on the pain in the Rheumatoid Arthritis (RA) in this review, it may be good if authors could add some description of the connection between JAK-mediated pain and RA in the text.

Author Response

Dear reviewer, thank you for your precious time, thank you for your review and fruitful demands for corrections.

Bellow are my point-to-point response to your required corrections:

Specific comment

  1. Since some paragraph repeats the similar message (e.g., lines 132-149 and lines 192-197), authors had better revise the manuscript by reconfiguration.
    Answer: I checked again the text and both of the paragraphs are too similar, I wanted to give different perspective on RA pain, being different from pain from all rheumatic diseases but ended up giving relatively same information.
  2. Authors should add appropriate references in the text. For example, there are no references in the text lines 162-190, lines 207-223).
    Answer: new references are added and they are properly numbered in the narrative 
  3. Authors had better add some figures or schemes that explain the JAK-mediated mechanism of pain for example for easier understanding for the reader.
    Answer: Thank you very much for this, figures always make an article better looking. I have added figure.
  4. Although authors focused on the pain in the Rheumatoid Arthritis (RA) in this review, it may be good if authors could add some description of the connection between JAK-mediated pain and RA in the text.
    Answer: I have added a couple of paragraphs regarding JAK-mediated pain and RA.               
    All the changes made for your demands are with yellow, the required changes for RA and JAK-mediated pain are with green for easier check.

Round 2

Reviewer 1 Report

The author did a good work improving the manuscript. In my opinion the paper can be accepted.

Author Response

Thank you for your revision!

Reviewer 2 Report

The manuscript has been improved.

I believe that it could benefit further from the addition of some original figures and tables. 

Author Response

Thank you again for your time! I have uploaded 2 figures and 1 table in the manuscript. I believe it got much better that way! This also lead to one additional reference!

All relevant changes I have marked with purple!

Reviewer 3 Report

Authors appropriately responded to my comments. I think the revised manuscript will be suitable for publication in the Life.

Author Response

Thank you for your fruitful help!